# Children with Fetal Alcohol Syndrome (FAS): 3D-Analysis of Palatal Depth and 3D-Metric Facial Length

**DOI:** 10.3390/ijerph17010095

**Published:** 2019-12-21

**Authors:** Moritz Blanck-Lubarsch, Dieter Dirksen, Reinhold Feldmann, Cristina Sauerland, Ariane Hohoff

**Affiliations:** 1Department of Orthodontics, University Hospital Münster, Albert-Schweitzer-Campus 1, 48149 Münster, Germany; hohoffa@uni-muenster.de; 2Department of Prosthodontics and Biomaterials, University Hospital Münster, Albert-Schweitzer-Campus 1, 48149 Münster, Germany; dieter.dirksen@ukmuenster.de; 3Department of Pediatrics, University Hospital Münster, Albert-Schweitzer-Campus 1, 48149 Münster, Germany; feldrei@uni-muenster.de; 4Institute of Biostatistics and Clinical Research, University of Münster, Schmeddingstraße 56, 48149 Münster, Germany; christina.sauerland@ukmuenster.de

**Keywords:** fetal alcohol syndrome (FAS), fetal alcohol spectrum disorder (FASD), 3D facial scan, palatal depth, vertical facial dimensions

## Abstract

*Background*: Drinking alcohol during pregnancy can result in severe developmental disorders in the child. Symptoms of the fetal alcohol spectrum disorder (FASD) comprise growth deficiencies, abnormal facial phenotype and damage or dysfunction of the central nervous system. Numerous diagnostic methods for facial phenotyping in FASD exist, but diagnoses are still difficult. Our aim was to find additional and objective methods for the verification of FAS(D). *Methods*: Three-dimensional dental models of 60 children (30 FAS and 30 controls) were used to metrically determine maximum palatal depths at the median palatine raphe. Three-dimensional facial scans were taken, and vertical distances of the face were measured at five defined facial landmarks (FP1–FP5) for each child. *Results*: Mean palatal height, total facial length (FP1–FP5) as well as FP4–FP5 did not significantly differ between the FAS group and the control group. Comparing vertical facial subdivisions, however, resulted in significant differences for distances FP1 to FP2 (*p* = 0.042, FAS > controls), FP2 to FP3 (*p* < 0.001, FAS < controls), FP3 to FP4 (*p* < 0.001, FAS > controls) and FP3 to FP5 (*p* = 0.007, FAS > controls). *Conclusions*: Metric vertical measurements of the face can be used as additional objective criteria for FAS diagnoses. However, no significant differences were reported for palatal depth evaluation in the specific age range tested in the present study.

## 1. Introduction

Fetal alcohol spectrum disorder (FASD) is an umbrella term used for developmental disorders in a foetus caused by maternal alcohol intake during pregnancy [1]. The consequences of this disorder can be severe and may result in high health care costs [2,3]. The symptoms of FASD are highly variable and comprise severe growth deficiencies, abnormal facial phenotype and damage or dysfunction of the central nervous system. According to Popova et al., a correlation between FASD and 428 accompanying diseases is possible [4]. Factors such as maternal age, smoking, nutrition status or variability in metabolism and genetic background may have an additional impact on the teratogenic effect of alcohol intake [5,6]. According to a study, the timing and extent of alcohol exposure may also have an effect on the occurrence of FASD in the child [7,8].

Fetal alcohol syndrome (FAS) is the most severe form of FASD, followed by partial fetal alcohol syndrome, alcohol-related birth defects and alcohol-related neurodevelopmental disorder [9].

FASD prevalence shows regional variability, with the highest prevalence in the European region, with 19.8 per 1000 births, and the lowest in the eastern Mediterranean countries, with 0.1 per 1000 [5]. For individual countries, the highest prevalence was found in South Africa, with 111.1 per 1000 births, followed by Croatia with 53.3 per 1000 and Ireland with 47.5 per 1000 [5]. The overall worldwide prevalence is estimated to be 7.7 per 1000 births [5]. According to a meta-analysis by Popova et al., the global prevalence of alcohol intake during pregnancy was estimated to be 9.8 per 100, and the prevalence of the most severe form, FAS, in the general population was estimated to be 14.6 per 10,000 [10].

In clinical practice, the early and correct diagnosis of FASD is important. At present, the four-digit diagnostic code, which was introduced by Astley and Clarren in 2000, is the most commonly-used diagnostic tool [11,12,13]. Its evaluation consists of the following four components: growth deficiency, facial phenotype, central nervous system damage or dysfunction and gestational exposure to alcohol [11,12,13].

Since patients are more regularly seen by dental and orthodontic specialists than by general practitioners, new noninvasive methods of diagnosing characteristic facial and oral phenotypes for FAS(D) are giving rise to the possibility of early FASD detection. Furthermore, most available diagnostic methods are, in parts, subjective. Noninvasive 3D scanning methods of facial and intraoral structures are becoming increasingly popular in dental and orthodontic practice. These scans enable extremely accurate measurements in all dimensions [14,15]. In addition, a 3D scan is easy to conduct, and can therefore be performed cost-efficiently by nonmedical staff. It is also quick, comfortable to the patient and allows objective measurements to be made; therefore, it could be a valuable asset in FAS(D) diagnoses. In recently published studies by Blanck-Lubarsch et al., metric philtrum depth determined by 3D facial scans could serve as an objective diagnostic indicator to aid and confirm FAS diagnoses [16]. Craniofacial measurements obtained using 3D extraoral scanners showed excellent reliability and accuracy, which makes this method applicable for clinical and scientific use [17]. Additionally, intraoral powder-free scanners showed equal precision and significantly higher patient comfort when compared with conventional impression techniques [18]. Dental arch dimensions showed deficiencies of the maxillary complex in the transversal dimension, resulting in a higher prevalence of crossbites in children with FAS [19]. Developmental defects of enamel and decayed, missing, filled teeth indexes were significantly higher for children with FAS, and significant differences in facial parameters such as profile, eye, nose and composition of facial thirds could be found [20,21,22].

These findings led to the assumption that further intraoral maxillary and extraoral vertical peculiarities might be found in children with FAS. 

Therefore, the aim of this study was to measure intraoral and extraoral vertical dimensions of the maxilla and face, respectively, using a 3D scan methodology to establish further diagnostic parameters characteristic for FAS(D), thus enabling the early recognition, and promotion of mental and physical development, of children with FAS(D).

## 2. Materials and Methods 

### 2.1. Study Design, Setting and Participants

In this prospective, cross-sectional study, 30 children with confirmed FAS diagnosis (FAS group) and 30 healthy children without FAS (control group) were examined; all the participants were of Caucasian origin. Children with FAS were recruited from 2012 to 2016 by a specialist in the Pediatric Department of the University Clinic Münster, who introduced our study to patients diagnosed with FAS according to the German FAS diagnostic guidelines [1]. The control group consisted of children from local schools prospectively included and examined during the same time period. The 3D facial scans as well as examinations of both groups were done by a trained orthodontic specialist in the Orthodontic Department of the University Hospital Münster. Inclusion criteria were mixed dentition for both groups and verified or refuted FAS diagnosis for the FAS and control groups, respectively. Exclusion criteria were primary or permanent dentition, completed or current orthodontic treatment as well as the presence of any disorder, disease or syndrome other than FAS influencing craniofacial or intraoral phenotype.

### 2.2. Variables and Data Sources/Management

For intraoral 3D assessments of palatal depth, alginate impressions (Cavex, ColorChange FS, Norden, Germany) of the upper jaw were taken, and orthodontic plaster models were created (Heraeus Kulzer, Moldano hard plaster type 3, Hanau, Germany), which were then scanned three-dimensionally (Atos, GOM GmbH, Braunschweig, Germany). With an analysis program (Atos Professional—3D Scanning and Inspection Software V8, GOM GmbH, Braunschweig, Germany), the palatal depth of each model was quantified, as shown in Figure 1a–c. This method was modified for 3D digital models in accordance with measurements of palatal depth as described by Korkhaus et al. for plaster models [23]. A similar method for 3D digital models was used by Gan et al. [24]. A horizontal plane (p1) was defined based on three reproducible landmarks, namely, the most prominent point of the Papilla Inzisiva (PP1) and the intersection of the palatal groove of the right (PP2) and left (PP3) first molar with the gingiva (Figure 1a). Perpendicular to this horizontal plane, a second plane (p2) was constructed median-sagittally based on the visible anterior–posterior course of the median palatine raphe (Figure 1b). The intersection line of this second plane in the plaster model allowed measurements of palatal depth to be taken at any point along the median palatine raphe perpendicular to the first plane (p1) with the maximum distance used for further analysis (Figure 1c).

The metric length measurements of the face in the vertical dimension were obtained from 3D surface reconstructions of the face created by a photogrammetry-based, contact-free optical method for capturing facial structures. Three-dimensional images were taken according to a standardised protocol with a defined distance and orientation of the face, defined by light beam localisers for the bipupillary line, the vertical facial midline perpendicular to the bipupillary line and the Frankfort horizontal line. The facial scanning device and scanning method were developed at the University Hospital Münster [25]. The facial scanning device consisted of three charge-coupled device cameras (resolution 1024 × 768 pixels, Imagingsource GmbH, Bremen, Germany) mounted on a horizontal track with a digital interface (IEEE1394), with the two outer cameras recording monochrome and the central camera collecting RGB colour images. A sequence of 13 different fringe patterns, projected onto the facial surface by an LCD projector (VT 58, NEC), was recorded by each camera of the system within 1 s, resulting in a point cloud of about 50,000–800,000 coordinates [25]. The individual coordinate points were then connected via Delaunay triangulation [26] for calculation of a 3D facial surface, and the colour information of the central camera was added.

Further analysis of the 3D data was done with the program gview (developed at the University Hospital Münster). Each point of the face was defined in a 3D coordinate system, which allows metric measurements to be made. For the identification of total facial length and its subdivisions, five points, i.e., FP1–FP5, which were easy and reliable to pinpoint in 3D photogrammetry [15] were defined within the median-sagittal plane from a frontal view of the face: FP1 (transition point of the hairline to the forehead), FP2 (the central point between the eyebrows just above the nose), FP3 (transition point from the nose to the upper lip), FP4 (transition point between the upper and the lower lip) and FP5 (most caudal point of the chin). The vertical distances between the respective points FP1–FP5 were measured and compared between the groups (Figure 2).

All scans and measurements were performed by the same experienced orthodontist. Prior to measurements and statistical evaluation, the orthodontist was blinded to all data regarding study groups.

### 2.3. Statistical Analysis

All analyses were performed with the software IBM^®^ SPSS^®^ Statistics 25 (IBM, Armonk, NY, USA). Metric variables were characterised by the arithmetic mean (M), standard deviation (SD), median (MD) and range (minimum, maximum). Due to violations of the requirements for parametrical testing (Kolmogorov–Smirnov test), the Mann–Whitney U test was used to assess the differences between FAS and control groups, whereas Fisher’s exact test was employed to evaluate the possible differences in gender distribution. To determine the intrarater reliability, Cronbach’s alpha was used. All analyses were regarded as explorative, and p-values were interpreted descriptively. Therefore, no adjustment for multiple testing was made. The local two-sided significance level p was set at 5%.

## 3. Results

### 3.1. Study Participants

The study groups consisted of 30 participants each. In the FAS group, 15 patients were male and 15 were female, whereas the control group consisted of 12 female and 18 male children. The average age of the participants was 8.5 years (SD 1.6), with an average of 8.8 years (SD 1.5) for the FAS group and 8.2 years (SD 1.8) for the control group (Table 1).

### 3.2. Main Results

Mean palatal depth was slightly higher for the FAS group (M ± SD = 12.6 ± 1.5 mm) as compared to the control group (M ± SD = 12.5 ± 1.7), but did not differ significantly (*p* = 0.708) (Table 1).

Measurements for total facial length (FP1–FP5) were not significantly different when comparing FAS (M ± SD = 151.1 ± 9.0 mm) and control group (M ± SD = 151.8 ± 8.1 mm) measurements (*p* = 0.737) (Table 1). However, comparing the facial subdivisions resulted in significant, and in parts highly significant, differences.

The vertical distance between FP1 and FP2 differed significantly (*p* = 0.042), with the FAS group showing a higher mean distance than the control group (53.4 ± 4.5 mm vs. 50.8 ± 4.2 mm, M ± SD) (Table 1).

For the distance FP2 to FP3, a highly significant difference could be found with *p* < 0.001, and the mean distance was lower for children with FAS (42.6 ± 3.6 mm vs. 49.1 ± 3.1 mm, M ± SD) (Table 1).

Measurements of FP3 to FP4 showed highly significant differences (*p* < 0.001), with greater mean distance in the FAS group compared to the control group (19.6 ± 2.3 mm vs. 16.5 ± 1.9 mm, M ± SD) (Table 1).

The vertical distance between FP3 and FP5 was significantly different between the groups (*p* = 0.007). In this case, the mean distance was higher for the FAS group (55.1 ± 4.4 mm vs. 51.9 ± 3.7 mm, M ± SD) (Table 1).

The distance FP4 to FP5 did not differ significantly between the groups (*p* = 0.701) (Table 1).

The conformity of measurements was high, with an intrarater reliability of 0.944 for FP1–FP2, 0.903 for FP2–FP3, 0.972 for FP3–FP5 and 0.950 for total facial length FP1–FP5.

## 4. Discussion

To minimise bias, children in the control group were recruited from local schools, instead of the Orthodontic University Department, avoiding a selection of extreme malocclusions and oral phenotypes, which could possibly have influenced palatal and facial contour.

The evaluation of facial parameters in FAS(D) diagnostics is currently based in part on visual, and thus subjective, assessment. The heterogeneity and fading of visual parameters in the later stages of life make the diagnosis of FAS(D) extremely challenging, leaving many patients without adequate health care and developmental support [27]. To date, the most commonly-used tool for the evaluation of facial features in FASD patients is the lip-philtrum guide proposed by Astley et al., which is based on a visual comparison of the lips, philtrum and midfacial development to photographs [28].

The aim of our study on children with FAS was finding new parameters and providing additional objective measures for early recognition of FAS(D).

Our study showed that for patients with an average age of 8.5 years, palatal height measurements may not be suitable for verification of FAS(D) diagnostics, since no significant differences between the groups could be detected. The impression of a high arched palate in patients with FAS, which is described in a study by Jackson et al. [29], might therefore be subjective in nature and a visual illusion because of transversal underdevelopment of the maxillary complex, which can be found in patients with FAS [19]. This phenomenon, i.e., that the visual impression of the vertical dimension can be dependent on the transverse dimension, has been described in the literature [30,31]. However, studies suggest that palatal growth and change in palatal depth may be age-dependent [32]. Therefore, further studies should be conducted to assess the palatal depth of children of different ages with FAS.

Vertical measurements of the face based on 3D facial scans, however, showed significant differences in all subdivisions with the exception of FP4 to FP5, and thus clarified that children with FAS have abnormal vertical proportions, with an emphasis on FP2 to FP3, which was significantly shorter, and the distance FP3 to FP4, which was significantly longer than in the control group. The measurement FP3 to FP4, in particular, represents philtrum length, which, therefore, can be expected to be longer in children with FAS. Comparing our findings with philtrum growth charts by Zankl et al. [33] underlines the fact that mean philtrum length in children with FAS is located around two standard deviations above normal values. The measurements for FP3 to FP4 in the control group were in accordance with the mean values on the philtrum growth charts by Zankl et al., although the landmarks for philtrum length by Zankl et al. were shown to be slightly different [33]. This shows that the philtrum is not only shallower, but also abnormal in the vertical dimension in patients with FAS [16]. The shorter distance of FP2 to FP3 hints at 3D underdevelopment of skeletal structures in the maxillary region, a theory that is supported by the higher prevalence of lateral crossbites in children with FAS [19].

Measurements of facial and intraoral anatomy based on 3D scans make highly-accurate evaluations possible of all structures with just one scan. Since 3D scans work without radiation, are contactless and can be performed within a short time period, they are comfortable and easily accepted by patients. Furthermore, they can be performed cost-efficiently by medical staff. With further technical advancements, 3D scans are becoming increasingly popular for clinical and private use, which could be an advantage in future diagnostic processes concerning patients with FAS(D). Using 3D scans, the measuring methods presented in this study could be easily applied by clinicians, and a worldwide data bank on FAS(D) measurements could easily be established.

In our study, patients with FAS, being the most severe form of FASD, were exclusively included. In the future, it might therefore be interesting to perform further studies on patients with milder symptoms of FASD to investigate possible correlations with, or deviations from, the findings in patients with FAS.

## 5. Strength and Limitations

This study was limited to Caucasian patients in mixed dentition with an average age of 8.5 years. At this point, craniofacial structures are still growing, which is why further studies should assess possible significant aberrations at earlier and later ages and growth stages. In addition, the results should be verified for other ethnic groups. Another limitation could be the accessibility to scanning devices. However, 3D scanners are becoming increasingly popular and will certainly become the gold standard in diagnostics in the future because of their scan time, reliability and accuracy [17,18].

A strength of our study was that measurements were performed metrically, thus enabling comparisons to be made and the development of reference values for diagnostic purposes. The usage of 3D scanning and 3D values makes our results highly reliable.

## 6. Conclusions

Palatal depth evaluation did not show any significant differences in children with FAS compared to healthy controls. Vertical facial measurements, however, proved to be suitable as additional objective parameters for FAS(D) diagnosis, with the distance of the midfacial length (FP2 to FP3) being significantly shorter and the distance of the philtrum length (FP3 to FP4) significantly longer in children with FAS. We therefore conclude that lower values for the midfacial length (distance FP2 to FP3) and higher values for philtrum length (distance FP3 to FP4) can be expected in children with FAS(D). The findings in our study can therefore contribute as additional metrics, and thus objective parameters, in FAS(D) diagnostics.

## Figures and Tables

**Figure 1 ijerph-17-00095-f001:**
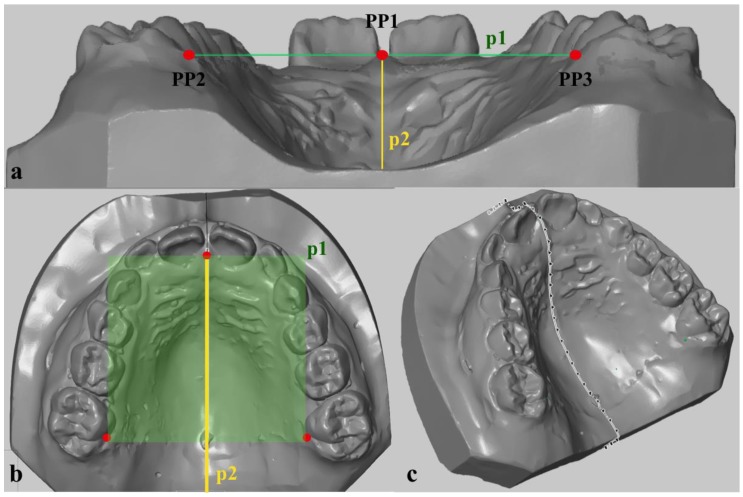
(**a**,**b**): Quantification of palatal depth: Horizontal plane (p1), defined by the Papilla Inzisiva (PP1), and the intersection of the palatal groove of the right (PP2) and left (PP3) first molar with the gingiva. Second plane (p2) perpendicular to p1 constructed median-sagittally along the median palatine raphe. (**c**): The intersection line of p2 with the plaster model allowed measurements of palatal depth to be taken along the median palatine raphe perpendicular to p1. The maximum distance of each plaster model was used for further analysis.

**Figure 2 ijerph-17-00095-f002:**
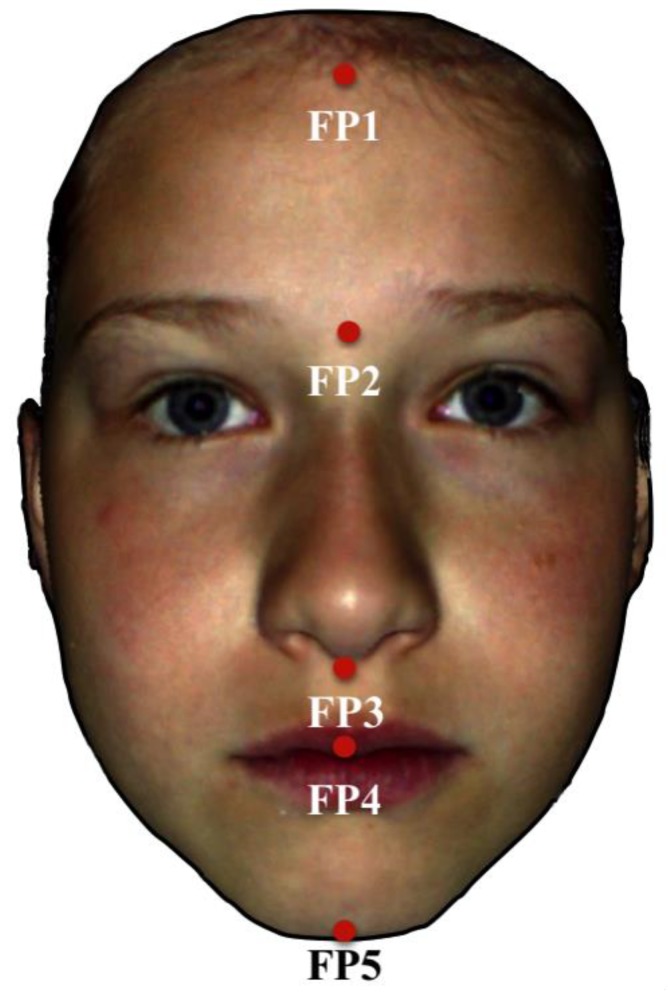
Three-dimensional facial surface reconstruction showing the landmarks FP1 to FP5 used for vertical extraoral facial analysis. FP1 = Transition point of the hairline to the forehead, FP2 = central point between the eyebrows just above the nose, FP3 = transition point from the nose to the upper lip, FP4 = transition point between the upper and the lower lip and FP5 = most caudal point of the chin.

**Table 1 ijerph-17-00095-t001:** Descriptive and analytical statistics for all outcome parameters evaluated. ^1^ Mann–Whitney U test; ^2^ Fisher’s exact test.

Investigated Parameters	Total	FAS Group	Control Group	*p*-Value
Gender				0.604 ^2^
Male	33	15	18	
Female	27	15	12	
Age at examination, years				0.095 ^1^
Mean (SD)	8.5 (1.6)	8.8 (1.5)	8.2 (1.8)	
Median (Range)	8.3 (5.8–11.9)	8.6 (6.6–11.2)	7.6 (5.8–11.9)	
Palatal depth, mm				0.708 ^1^
Mean (SD)	12.6 (1.6)	12.6 (1.5)	12.5 (1.7)	
Median (Range)	12.4 (9.3–16.8)	12.6 (10.1–16.8)	12.3 (9.3–16.2)	
Total facial length (FP1–FP5), mm				0.737 ^1^
Mean (SD)	151.5 (8.5)	151.1 (9.0)	151.8 (8.1)	
Median (Range)	153.6 (127–168.4)	153.5 (133.8–168.4)	154.2 (127–163.9)	
FP1–FP2, mm				0.042 ^1^
Mean (SD)	52.1 (4.5)	53.4 (4.5)	50.8 (4.2)	
Median (Range)	52 (42.75–63.2)	54.2 (46.3–63.2)	50.8 (42.8–57)	
FP2–FP3, mm				<0.001 ^1^
Mean (SD)	46 (4.6)	42.6 (3.6)	49.1 (3.1)	
Median (Range)	46.2 (36.2–54.3)	41.5 (36.2–50.8)	49.6 (41.2–54.3)	
FP3–FP4, mm				<0.001 ^1^
Mean (SD)	18 (2.6)	19.6 (2.3)	16.5 (1.9)	
Median (Range)	18 (12.4–24.9)	19.8 (14.8–24.9)	16.7 (12.4–18.9)	
FP3–FP5, mm				0.007 ^1^
Mean (SD)	53.4 (4.3)	55.1 (4.4)	51.9 (3.7)	
Median (Range)	53.2 (43.1–65.9)	56.2 (45.5–65.9)	51.9 (43.1–60.2)	
FP4–FP5, mm				0.701 ^1^
Mean (SD)	35.6 (3)	35.7 (3.1)	35.5 (2.9)	
Median (Range)	35.3 (29.8–42.3)	36 (30.3–41.8)	34.7 (29.8–42.3)

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
