# Peer review of "Children with Fetal Alcohol Syndrome (FAS): 3D-Analysis of Palatal Depth and 3D-Metric Facial Length"

_ijerph, 2019, doi:10.3390/ijerph17010095_

Round 1
Reviewer 1 Report
Dear Authors,
I have read with interest the manuscript and some questions raised. Attached please find my comments.
Overall. General English grammar revision (Minor spelling errors).
Abstract: Authors stated “palatal depth is not a suitable parameter”. This sentence is a bit strong, as differences could be found in different ages. It could be rephrased as “no significant differences were reported after palatal depth evaluation in the specific age range tested in the present report”.
Introduction. Authors stated that “In addition, a 3D-scan is easy to apply and can therefore be performed cost-efficiently by non-medical staff. It is also fast and comfortable for the patient and allows objective measurements, which could be a valuable asset in FAS(D) diagnosis”. Authors could stress more the importance of precision and comfort these recently introduced devices with adequate references, dividing facial and intraoral scanning systems. It could be added that “Craniofacial measurements obtained with 3D extraoral scanners showed excellent reliability and accuracy, which qualifies this method for clinical and scientific use (Franco de Sá Gomes C, Libdy MR, Normando D. Scan time, reliability and accuracy of craniofacial measurements using a 3D light scanner. J Oral Biol Craniofac Res. 2019 Oct-Dec;9(4):331-335). Additionally, intraoral powder free scanners showed equal precision and significantly higher patient comfort if compared with conventional impression technique (Computerized Casts for Orthodontic Purpose Using Powder-Free Intraoral Scanners: Accuracy, Execution Time, and Patient Feedback. Sfondrini MF, Gandini P, Malfatto M, Di Corato F, Trovati F, Scribante A. Biomed Res Int. 2018 Apr 23;2018:4103232)”.
Materials and Methods. Authors stated “In this prospective, cross-sectional study, 30 children with confirmed FAS diagnosis and 30 healthy children without FAS (control group) were examined, all 60 of Caucasian origin”. Please add if and how sample size has been calculated.
Materials and Methods. Authors stated “the palatal depth of 100 each model was quantified as shown in Figure 1”. Please add a reference for this method.
Materials and Methods. Authors stated “Due to violations of requirements for parametrical testing”. Please add the name of statistical test used to assess the normality of the distributions.
Discussion. Authors stated “Our study could show that palatal height measurements are not suitable for verification of FAS(D) diagnostics, since no significant differences between the groups could be detected”. Patients are growing subjects (8.5 years (SD 1.6)). Therefore, the validity of the results is reliable only in this specific age range. It is not known if further differences could be found during peak growth spurt (11-13 years). Therefore, this important concern should be pointed out in discussion section.
Discussion. Authors could point out at the end of the paragraph a couple of rows about limitations of the present report (for example, but not limited to: no 3D acquisition over time, but in a single age…).
References. Some references are quite old (1999,1995). Where possible please switch with some more modern research.
Figures: ok
Tables: ok.
Author Response
Dear Reviewer,
we would like to thank you for your very detailed and helpful comments. The manuscript was revised and corrected according to the comments. We highlighted our corrections in the manuscript with green colour.
Reviewer 1:
1) Abstract: Authors stated “palatal depth is not a suitable parameter”. This sentence is a bit strong, as differences could be found in different ages. It could be rephrased as “no significant differences were reported after palatal depth evaluation in the specific age range tested in the present report”.
Answer: Thank you for your comment. We changed the sentence as you suggested.
2) Introduction. Authors stated that “In addition, a 3D-scan is easy to apply and can therefore be performed cost-efficiently by non-medical staff. It is also fast and comfortable for the patient and allows objective measurements, which could be a valuable asset in FAS(D) diagnosis”. Authors could stress more the importance of precision and comfort these recently introduced devices with adequate references, dividing facial and intraoral scanning systems. It could be added that “Craniofacial measurements obtained with 3D extraoral scanners showed excellent reliability and accuracy, which qualifies this method for clinical and scientific use (Franco de Sá Gomes C, Libdy MR, Normando D. Scan time, reliability and accuracy of craniofacial measurements using a 3D light scanner. J Oral Biol Craniofac Res. 2019 Oct-Dec;9(4):331-335). Additionally, intraoral powder free scanners showed equal precision and significantly higher patient comfort if compared with conventional impression technique (Computerized Casts for Orthodontic Purpose Using Powder-Free Intraoral Scanners: Accuracy, Execution Time, and Patient Feedback. Sfondrini MF, Gandini P, Malfatto M, Di Corato F, Trovati F, Scribante A. Biomed Res Int. 2018 Apr 23;2018:4103232)”.
Answer: Thank you for your suggestion. We added the sentences and references as you recommended.
3) Materials and Methods. Authors stated “In this prospective, cross-sectional study, 30 children with confirmed FAS diagnosis and 30 healthy children without FAS (control group) were examined, all 60 of Caucasian origin”. Please add if and how sample size has been calculated.
Answer: As pointed out in section 2.3 (statistical analysis), the study was exploratory in nature and therefore no statistical power has been estimated.
4) Materials and Methods. Authors stated “the palatal depth of each model was quantified as shown in Figure 1”. Please add a reference for this method.
Answer: Thank you for your suggestion. The method for palatal depth measurement was modified in accordance with a method described by Korkhaus et al.. We included two references for clarification..
5) Materials and Methods. Authors stated “Due to violations of requirements for parametrical testing”. Please add the name of statistical test used to assess the normality of the distributions.
Answer: Thank you for our comment. We used Kolmogorov-Smirnov test and clarified this in the statistical analysis section (2.3).
6) Discussion. Authors stated “Our study could show that palatal height measurements are not suitable for verification of FAS(D) diagnostics, since no significant differences between the groups could be detected”. Patients are growing subjects (8.5 years (SD 1.6)). Therefore, the validity of the results is reliable only in this specific age range. It is not known if further differences could be found during peak growth spurt (11-13 years). Therefore, this important concern should be pointed out in discussion section.
Answer: Thank you for your comment. We clarified this in the discussion section.
7) Discussion. Authors could point out at the end of the paragraph a couple of rows about limitations of the present report (for example, but not limited to: no 3D acquisition over time, but in a single age…).
Answer: We included a paragraph concerning strengths and limitations of our study as you suggested.
8) References. Some references are quite old (1999,1995). Where possible please switch with some more modern research.
Answer: We included a more recent reference on the topic of the study from 1999. Concerning the reference from 1995 we are very sorry, but this is the original study describing the method of the lip-philtrum guide.
Reviewer 2 Report
This observational study compared 30 children in the mixed dentition with FAS and 30 non-syndromic children from local schools. Palatal and facial measurements were taken from maxillary dental models and 3D-photogrammetry. Facial measurements (facial heights) were statistically different in children with FAS compared to controls. On the other hand, no differences were found for palatal measurements. The study is new and clinically useful defining better the phenotype of FAS.
Some minor suggestions are given:
In the title, FAS should come first instead the methodology used for evidence on the irregularity instead of on the method. Statistical power should be given What is the reliability of the method used? Authors performed the study error analysis? In the discussion, line 224, authors should be careful to associate vertical and transversal deficiency of the maxilla. Not all patients with vertical deficiency of maxilla has transversal deficiency and posterior crossbite. Conclusion should state the name facial height instead of the name of measurement (FP1-FP2).Author Response
Dear Reviewer,
we would like to thank you for your very detailed and helpful comments. The manuscript was revised and corrected according to the comments. We highlighted our corrections in the manuscript with yellow colour.
Reviewer 2:
1) In the title, FAS should come first instead the methodology used for evidence on the irregularity instead of on the method.
Answer: Thank you very much for your comment. We changed the order of the title as you recommended.
2) Statistical power should be given What is the reliability of the method used?
Answer: As pointed out in section 2.3 (statistical analysis), the study was exploratory in nature and therefore no statistical power has been estimated. We included two references in the manuscript concerning accuracy and reliability of 3D-scanning devices (please see yellow highlights).
3) Authors performed the study error analysis?
Answer: Using the analysed parameters (FP1-FP2; FP2-FP3; FP3-FP5) all patients could be diagnosed correctly, if only one measurement, for example FP2-FP3 is used the area under the ROC-curve was 89,6 % as shown in the ROC-curve in the attachment.
4) In the discussion, line 224, authors should be careful to associate vertical and transversal deficiency of the maxilla. Not all patients with vertical deficiency of maxilla has transversal deficiency and posterior crossbite.
Answer: Thank you for your comment. In this section, we meant to explain, that even though the vertical dimension of the palate is not underdeveloped, there might be a visual impression of a high palate if the maxilla is underdeveloped in the transverse dimension. We clarified in the manuscript that this phenomenon is described in literature as the illusion of the vertical-horizontal dimension and included references.
5) Conclusion should state the name facial height instead of the name of measurement (FP1-FP2).
Answer: Thank you for your suggestion. We included the names of the measuring points for each distance for clarification. Please see yellow highlights in the manuscript.

Round 2
Reviewer 1 Report
Dear Authors,
thank you for revising your manuscript.